# Algorithms and Faith: The Meaning, Power, and Causality of Algorithms in Catholic Online Discourse

## Radosław Sierocki

Faculty of Social Sciences, University of Warmia and Mazury in Olsztyn, 10-719 Olsztyn, Poland; radoslaw.sierocki@uwm.edu.pl

**Abstract:** The purpose of this article is to present grassroots concepts and ideas about "the algorithm" in the religious context. The power and causality of algorithms are based on lines of computer code, making a society influenced by "black boxes" or "enigmatic technologies" (as they are incomprehensible to most people). On the other hand, the power of algorithms lies in the meanings that we attribute to them. The extent of the power, agency, and control that algorithms have over us depends on how much power, agency, and control we are willing to give to algorithms and artificial intelligence, which involves building the idea of their omnipotence. The key question is about the meanings and the ideas about algorithms that are circulating in society. This paper is focused on the analysis of "vernacular/folk" theories on algorithms, reconstructed based on posts made by the users of Polish Catholic forums. The qualitative analysis of online discourse makes it possible to point out several themes, i.e., according to the linguistic concept, "algorithm" is the source domain used in explanations of religious issues (God as the creator of the algorithm, the soul as the algorithm); algorithms and the effects of their work are combined with the individualization and personalization of religion; algorithms are perceived as ideological machines.

**Keywords:** algorithm; authority; Catholic discourse; folk theories; online forum; Roman Catholic Church

## 1. Introduction

### 1.1. Social Meanings of Algorithms

We live in an algorithmic society (Burrell and Fourcade 2021; Lash 2007; Schuilenburg and Peeters 2021), and the influence of algorithms extends to diverse facets, notably affecting religion, religious thinking, the religious worldview, and the very concepts of religion, including vernacular theories of religion. In this paper, I argue that the influence relies not only on the actual work of the algorithms, but also on the meanings that are ascribed to algorithms, and—from a wider perspective—to artificial intelligence and on attempts to understand their roles in contemporary society.

Algorithms help make decisions, often taking charge instead of users, operating discreetly in the background of our digital activities (Willson 2017, p. 139). The power and agency of algorithms derive from their role as code that shapes our society. This transformation results in a society characterized by "black boxes" and enigmatic technologies (Pasquale 2015, pp. 1–3). On the other hand, what proves more interesting in this article, is that their power emanates from the meanings we are inclined to attribute to algorithms, leading us to take certain actions and engage in social practices based on our imaginative perception of their power, despite their actual characteristics. From this perspective, algorithms evoke a broader notion of modern rationality and a certain worldview (Beer 2017, pp. 7–8). Nick Seaver points out that "there is no such thing as an algorithmic decision; there are only ways of seeing decisions as algorithmic" (Seaver 2018, p. 378). Sociologists would translate that into the theorem of William I. Thomas, that if people define the power and agency of algorithms as real, they are real in regard to their consequences.

The term "algorithm" is ambiguous. As a subject of interest for mathematicians, computer scientists, or programmers, the algorithm is a simple and non-controversial

thing; for a broader public that lacks a technical education, it is an incomprehensible and complex matter (Airoldi 2022, p. 7; Gillespie 2016, p. 18). The algorithms developed for social media, digital platforms, computer programs and games, and the products and services of technological giants, began to be identified as the processes of quantification and datafication of social reality and have permeated almost every aspect of social life. The category of algorithms is frequently discussed in the context of artificial intelligence, particularly concerning recent advancements, such as ChatGPT, a language model that operates based on algorithms. As such they became the subject of research by social scientists, who see them as the instruments of power and governance (Neyland and Möllers 2017, p. 45).

In the public consciousness, algorithms have evolved into mysterious artifacts, deemed impossible to comprehend or unravel (a black box, an enigmatic technology, an umbrella term), and the users of technology and the media are more eager to recognize the influence they exert and submit to their authority, rather than to attempt to understand them (Airoldi 2022, p. 37; Gillespie 2016, p. 18)

Tarleton Gillespie emphasizes that for computer scientists and engineers, an algorithm is a computational procedure enabling the achievement of a task in the simplest way. Algorithms are designed based on previously created models and then implemented in applications. Importantly, they are trained on datasets, which may have broader social implications than the algorithm itself, as these datasets can possess specific values, biases, or mismatches at odds with reality. In this context, algorithms are viewed as a "trick of the trade" (Gillespie 2016, pp. 19–22). In public discourse, the algorithm has become a synecdoche of information technologies treated as a set of data, training data, target effects, models, computational operations, applications and platforms, and subject users. The technical "independence" of algorithms is also usually assumed, forgetting that there are people involved at every stage of their work (Beer 2017, p. 5; Gillespie 2016, pp. 22–23). Massimo Airoldi makes use of the generic terms "algorithm" and "machine" interchangeably to "broadly indicate automated systems producing outputs based on the computational elaboration of input data" (Airoldi 2022, p. 8). The notion of an algorithm is also seen as a kind of talisman that allows it to disown responsibility and cede it to the technological systems taking decisions and performing actions. This applies both to technology companies and users (Gillespie 2016, p. 24).

From the perspective of social sciences, algorithms are not only lines of codes or the software facility, but rather the social processes that both synthesize and influence the social order. Ted Striphas, according to Raymond Williams, notes that the semantic dimension of algorithmic culture is as important as the technological one (Striphas 2015, p. 397). Many scholars point out that "algorithms" for computer sciences and social sciences are not the same. For social sciences, they are not singular, technical objects, but rather unstable and complex sociotechnical systems, constituted by social practices (Seaver 2017, s. 4–5; 2018, p. 378). It is also argued that it is not the algorithm that has the power and agency, but that the power is derived from "algorithm associations", which are "the assemblages of people, things, resources and other entities held together by practice and process", and the activity of algorithms depends on the situation in which they operate (Neyland and Möllers 2017, pp. 45–46). David Beer (2017, p. 2) stresses that "when thinking about the power of the algorithm, we need to think not just about the impact and consequences of code, we also need to think about the powerful ways in which notions and ideas about the algorithm circulate through the social world". In other words, the social power of algorithms depends both on things and words; hence, the need for analysis of the notions of algorithms and the discourse on algorithms (Beer 2017, p. 9).

The notions of algorithms are connected with power and omnipotence. Taina Bucher claims that "the power and politics of algorithms stem from how algorithmic systems shape people's encounters and orientations in the world". She adds that the power cannot be reduced to the code, but we should agree that algorithms are parts of hybrid assemblages with diffused personhood and relational agency. The power of algorithms is not the

"property of technology", but it is a question about the programmed sociality, and how it is "realised as a function of code, people, and context" (Bucher 2018, p. 8). According to Jan Kreft (2019, pp. 24–25), the power of an algorithm is its ability to impose a will, force a certain behavior, and control the actions of the user, who is a consumer and also a provider of data. Power objectifies the human being, although he or she is declaratively a subject, who makes certain choices. It is also its ability to impose a will on social groups and economic actors. These complex socio-technological systems blend the agency of humans and algorithms and lead to "distributed agency" (Beer 2017, pp. 4–5; Neyland and Möllers 2017, pp. 45–46).

The reconstruction of the algorithm (opening "the black box") would list the various entities connected to the algorithm association network (socio-technological system), including all human and non-human actors (Gutiérrez 2023; Latour 1991). As Rob Kitchin writes, the sociocultural research approaches to algorithms include (mentioning the most important) analyses of: the source code; the experiences of software programmers, designers, and coding teams; the input and output data; the institutional environment that algorithms are used in; and the experiences and opinions of users meeting algorithms in their everyday activities (Kitchin 2017, pp. 22–26). Among the methodologies of social research conducted on algorithms, the algorithmization of society, or datafication, the analysis of the meanings attributed to algorithms by ordinary users is one of the most interesting aspects of this paper. This is the research, as Beer writes, on the circulation of meanings attached to algorithms. We can assume that meanings have social consequences that are as important as the proper work of algorithms. Qualitative research dealing with grassroots, vernacular readings of algorithms derives from human–computer interactions research (DeVito et al. 2018; Eslami et al. 2016; Ridley 2023; Toff and Nielsen 2018) and uses a variety of methods: from text and discourse analysis to interviews with users. For example, the research by Ytre-Arne and Moe (2021) shows five folk theories based on open questions in a survey on algorithms. These folk theories point out that algorithms: (1) narrow the cognitive horizon instead of broadening it, because they lock the user into what he or she already knows and likes; (2) are practical, as they help select and sort information; (3) are reductive, because they operate based on stereotypical perceptions and representations; (4) are intangible, as their actions and effects are difficult to grasp and are non-transparent and, thus, arouse fears and suspicion, and they are associated with surveillance and a threat to free will; (5) are exploitative, as users feel that algorithms are collecting data about them for unclear and undisclosed purposes of the forces behind them.

*1.2. Algorithms in the Context of Religion*

Although the studies about religion and algorithms/AI in social sciences are not very developed yet, there is interesting research that is taking into account the issues of power and causality. In the context of religion, two frameworks for conceptualizing the algorithm can be outlined: atheistic and theistic. Beth Singler points out that the teleological, atheist narrative of rationality, the narrative about algorithms, is part of a larger narrative of modernity, rationalism, and secularization. In contrast, there are theistic narratives (intentional, parodic, metaphorical) about the deification of algorithms and artificial intelligence. In these interpretations, algorithms and AI have the characteristics of traditional and theological depictions of a monotheistic God: omnipotence, omniscience, and omnipresence, sometimes with omnibenevolence (Singler 2020, pp. 945–47). Randall Reed asks about the role of religion in the acceptance or resistance to increasing surveillance to which we are being subjected by the operations that use algorithms, artificial intelligence, and other technologies that track and record social decisions, behaviors, and practices. He refers to the meaning that Christianity ascribes to the doctrine about the all-seeing God and the influence of the feeling of being under constant surveillance for good behavior. Reed poses a question about replacing the omnipresence and omniscience of God with a conviction on the omnipresence and omniscience of Google and the potential opposition to entering God's territory (Reed 2021, pp. 1–2).

Singler analyzed the tweets containing the phrase: "Blessed by the algorithm" (BBtA). Although the phrase is not common, it is noteworthy because BBtA not only appears in parodistic tweets, but also in metaphoric expressions and tweets with ontological statements about reality. Second, the BBtA phrase shows the perception of the agency of algorithms and AI in public notions. The analysis shows that the expression is used in addition to the success of processes that depend on the algorithm's mood, i.e., when algorithms help with sharing the content on social media, provide interesting suggestions (i.e., Spotify), or are involved in beneficial tasks in the work. The phrase also indicates that the user generally favors luck and the phrase BBtA refers almost to a divine instance or providence. Some interpretations refer back to religion and assume that "algorithmic gods" direct the fate of the world. Users also complain that they were not blessed by the algorithm or were even cursed. There are also tweets alluding to concepts derived from science fiction, presenting the notion that since we can simulate universes (e.g., in computer games), we should assume that our world is also a simulation created by a higher intelligence. BBtA tweets are part of a theistic framework for understanding the algorithm (and AI) and present parodic–metaphorical notions of the algorithm as a causal force influencing the fate of users (Singler 2020, pp. 949–52).

Heidi Campbell writes—in the context of authority in digital culture—about "algorithmic authority". She claims, authority in this culture does not come from external sources and protocols (i.e., the authority of the Church), but from the media system itself. Values are extracted from information sources that are: "sorted and determined by non-human entities (i.e., search engines). The authority is based on rankings and reputation systems found online and depends on many likes, shares, comments, number of followers and visibility on Google searches" (Campbell 2021, pp. 8–9). Sasha A.Q. Scott is considering the inclusion of an application for confession and the consequences of replacing the priest with the silent logic of algorithmic automation for religious authorities (Scott 2016).

The topic of this article is the perception of algorithms by Catholics. The Roman Catholic Church takes a stance on new technologies in several ways, and although the main question in this paper is about the meanings of algorithms in vernacular, folk theories, an introduction on the position of the Church authorities towards algorithms and artificial intelligence is necessary.

The authorities of the Roman Catholic Church recognize the ongoing changes in the sphere of technology, understand the need to adapt to technological innovations, see the benefits, and acknowledge the risks, but it is the faithful who are more interested in incorporating new solutions and are quicker to test them than the institutional Church, which is a general feature of the institutions. The Church is cautious and critical of new technologies, as Cory Labrecque claims: "praises those technological interventions that have contributed to the well-being of humankind and the environment but expresses concern when human freedom is conflated with self-sufficiency and when the measure of human finality is the satisfaction of one's interests in the enjoyment of earthly goods. A self-sufficiency that attempts to eliminate our awareness of our dependence on God and fails to recognize human–human as well as human–nature interdependence falls short of the sort of mutual belongingness, faithfulness, and enduring responsibility" (Green et al. 2022, pp. 18–19).

The Roman Catholic Church draws attention to the technological development in Pope Francis' *Laudato Si*, and although the encyclical letter does not speak directly about algorithms, it refers to the social effects of technological development, which for decades has influenced, among others, the labor market, social exclusion, and inequalities in regard to access to energy. In addition, the omnipresent media and digital world impede learning about "how to live wisely, to think deeply and to love generously" (Pope Francis 2015, sec. 46–47).

One of the most important and frequently cited points in the Church's activities is the document *Rome Call for AI Ethics* (2020), signed by the Pontifical Academy for Life, Microsoft, IBM, the FAO, and the Italian Ministry of Innovation in Rome on 28 February

2020. The goal of the document is to promote an ethical approach to AI. *Rome Call* comprises three impact areas (education, ethics, rights) and six principles: transparency, inclusion, accountability, impartiality, reliability, security, and privacy.

*1.3. Research Questions*

Discussions by theologians and churchmen on the role, significance, opportunities, and dangers presented by the development of new technologies continue, pointing at the same time to the need to adapt the Church and its teachings to new situations, as well as to express caution about directly implementing new solutions based, for example, on algorithms into religious activities and practices. In this context, from the perspective of sociology, it is not only the discussions taking place among theologians, ethicists, and church hierarchs that are interesting, but also the grassroots discussions carried out by believers with neither theological nor IT training, or the discussions taking place, not in church congregational meetings or academic conferences, but in general spaces on the internet. Such discussions reveal commonly functioning folk theories about algorithms. Therefore, the main research question in this paper is how algorithms are understood, perceived, and imagined, and how their meanings are negotiated at the level of vernacular theories, represented by the faithful, belonging to the community of the Roman Catholic Church in Poland and participating in discussions on Catholic internet forums. The question posed in this way is broad and can be translated into a few detailed issues. First, how algorithms are understood. It is a question rather about the meanings and associations connected with "the algorithms", as in the discussions on religious forums talking or mentioning algorithms does not require giving a strict definition. The way of understanding algorithms can be reconstructed on the basis of more or less extensive statements that do not always refer directly to algorithms. It should be noted that religion is one of the many contexts in which such a vernacular understanding of algorithms may appear, and that the "religious" understanding may differ from the definition provided by programists and computer scientists. The question gets more complicated when the intensity of the discussions on artificial intelligence and, especially, ChatGPT increase on online forums. Any knowledge of the users about algorithms is not important for the operation of the algorithm itself and the understanding of the algorithm may be different for specialists and the user. When it comes to AI and language models, technology users participate in negotiations about the meaning by actively using it, i.e., ChatGPT, according to their ideas and religious convictions. It is also important to emphasize that there are more and more recommendations to take into account not only ethical, but also religious aspects in the development of AI (Fondazione Bruno Kessler 2021). In this paper "artificial intelligence" and "ChatGPT" were included in the analysis if they appeared in the discussions on algorithms and were used equivalently to "algorithm". We can assume that religious forums advocate integrating algorithms into ideological discussions, including questions about the theistic or atheistic framework and algorithms entering into God's place.

Second, how the understanding of algorithms in a particular context problematizes the issues related to agency and the choices made by algorithms against the background of social life, their influence on people's choices, and free will. It is also the question about meshing the human, the machine, and divine agency.

Third, how algorithms as the notions of meanings and as a part of complex socio-technical systems are connected with the understanding of authority, which is an issue related to the problem of agency. As has been said, for social sciences it is not the algorithm that is the issue, but the algorithmic associations that tie together lines of code, data, software engineers, users, IT knowledge, and popular ideas about what the algorithm is.

This research was based on analysis of posts that appeared in four Polish Catholic Forums: Wiara.pl (http://forum.wiara.pl), Katolik.pl (http://dyskusje.radiokatolik.pl/), Z Chrystusem.pl (https://zchrystusem.pl/), and Dolina Modlitwy (https://dolinamodlitwy.pl/forum/).

## 2. Materials and Methods

The methods used in this research are qualitative ones, and they can be included in netnography methods. Robert V. Kozinets (2010) understands netnography, like ethnography, as a cocktail (or bricolage) of methodologies: from participant observatories to descriptive statistics, and in this research discourse analysis comes to the fore. The sociological approach to discourse analysis assumes that discourse is a category that is independent of the texts that constitute it. Text is only the material representation of the discourse that is the social structure (Chalaby 1996, p. 688). Laclau and Mouffe (2001) claim that meanings are given in the specific system of differences. Any object may exist outside discourse, but it then cannot have any meaning. They write: "An earthquake or the falling of a brick is an event that certainly exists, in the sense that it occurs here and now, independently of my will. But whether their specificity as objects is constructed in terms of 'natural phenomena' or 'expressions of the wrath of God', depends upon the structuring of a discursive field" (2001, p. 108). Similarly, Howarth (2000) gives the example of the forest that might be an independent object but gets its meaning from discourse when it becomes an obstacle on a motorway, or the subject of special interest for ecologists and scientists. In this way, discursive structures are created, which constitute and organize social relations (2000, p. 102). In this paper, it is assumed that algorithms may function independently of the users' will and knowledge. However, within the field of sociology, they transform into "social objects" as they become the subject of social interest, conversations, and debates. This transformation occurs when users begin to form opinions and attitudes toward them.

This analysis is based on posts that appeared on four Polish Catholic online forums and mentioned the word "algorithm". The forums are: Wiara.pl ([W], 38.202 forum topics at the end of 2023), forum Katolik.pl ([K]; 5.042 forum topics), forum ZChrystusem.pl ([ZC]; 7.198 forum topics), and DolinaModlitwy.pl ([DM]; 1.308 forum topics). Wiara.pl (Faith) and Katolik.pl (Catholic) are the most popular Catholic forums in Poland. The forums are connected with Catholic portals and publishers: Wiara.pl was founded in 2001 and is part of the catholic weekly "Gość Niedzielny" (Katowice, Poland), and Katolik.pl was founded in 2000 by the Society for the Divine Saviour Foundation (Kraków, Poland). Forum ZChrystusem.pl (With Christ) appeared in 2015 as a non-institutional initiative, similarly Dolina Modlitwy (Valley of Prayer) appeared in 2020.

Posts on the forums are written according to the forum topics, which are ordered in thematic sections. All of the forums have administrators and moderators, who ensure compliance with the regulations. According to the regulations, Catholics, representatives of other denominations, and non-believers can post their statements, arguments, and opinions on the forums, while respecting Catholic values, and, usually, protestants, atheists, and agnostics take advantage of this opportunity to engage in polemics with Catholics, ask provocative questions, or wage "religious" wars.

The posts mentioning "algorithms" were scattered across various forum topics. There was no topic dedicated directly to the question about understanding algorithms in context of religion; therefore, the next part of this paper (results) is the reconstruction of the meanings that were ascribed to the category of "algorithm" in over 200 topics (discussions).

What is important for analyzing folk theories is that the content (posts) created on online forums (as well as on social media) does not require an ecclesiastical "imprimatur", and is not selected strictly by editors, as long as it is not inconsistent with the regulations. Most forum users are both home-grown theologians, creating or showing their understanding on the interpretation of the faith, and heretics, who are unaware of whether and how much their views, attitudes, and knowledge deviate from the official interpretation of the Church. Therefore, on internet forums, there are grassroots ideas about what algorithms (or AI) are capable of, and how they can be used for religious purposes, just as a dozen years ago, there were ideas about the ritual of confession or Mass via the internet or questions like can/do robots have souls?

Although online forums are losing popularity in favor of social media, the studied forums remain important places for the exchange of opinions, views, experiences, ideas,

questions, and answers for Catholic internet users in Poland (Kołodziejska 2020, p. 63). In comparison to social media platforms, which have dominated the social space of the internet when it comes to the number of users, internet forums seem to be more stable internet communities with loyal users, who are also more concentrated on discussions that engage rational and substantive argumentations. Some of the discussions continue for weeks or months, and they are less subject to the need to earn likes, shares, and other types of immediate rewards. The forums that are the subject of this study represent, on the one hand, the forums with a long history and traditions (Wiara.pl and Katolik.pl have been operating for over 20 years). On the other hand, the relatively new forums (ZChrystusem and Dolina Modlitwy) are examples of the ongoing demand for such spaces. Internet forums as communication platforms are to a lesser extent, and this is especially important in the context of the subject of this article, subordinate to or exposed to the operation of algorithms, which suggest specific content, topics, or friends.

The forum's content provides insights into common social perceptions of what algorithms are, how they work, or what they could be for the Church, religion, faith, and religious practices. Forums create archives of posts that are relatively easy to find, and search engines were used to find posts containing the word "algorithm" (in Polish: "algorytm"). Digital discourse analysis is a qualitative method, which is concentrated on rather small digital spaces, but some quantitative information about the topics, posts, and users should be provided in order to provide the background to the analyzed discourse (Recuber 2017, pp. 50–51). The following data were analyzed:

- One-hundred and fifty-two topics, 339 posts, 84 users (and an unidentified number of anonymous accounts) in regard to Wiara.pl; the first post appeared in 2004 (15 posts that year), the last one—in 2023 (10 posts that year); the most of posts mentioning "algorithm" appeared in 2015 (52 posts);
- Fifty-five topics, 193 posts, 58 users in regard to Katolik.pl; the first post appeared in 2010 (3 posts that year), the last one was in 2023 (29 posts that year); the most posts appeared in 2014 (32 posts);
- Twenty-eight topics, 42 posts, 24 users in regard to ZChrystusem; the first post appeared in 2017 (that was the only post that year), the last one was in 2023 (17 posts); the most posts appeared in 2023 (17);
- Four topics, 9 posts, 5 users in regard to Dolina Modlitwy; 4 posts appeared in 2022 and 5 in 2023.

Posts that refer directly to algorithms or barely mention them are scattered throughout the forums', in various topics and sections. Their analysis involved coding, which was both an effort to understand and reconstruct the points of view, as well as linking individual statements, data, and general ideas (Saldaña 2009, pp. 7–11).

In the case of the study of religion and technology, the study of online forums and social media is largely a search for signals of change within the study of certain technological/social trends, rather than looking for data that confirms the massiveness of the phenomena (see Dragt 2018; Kjaer 2014; Raymond 2010).

The algorithmization of society is undoubtedly one such broad techno-social trend, the question is how it manifests within religion. It seems that we would sooner find these changes in the social practices and rituals performed on the private laptops and smartphones of the networked faithful, than in the Church treated as an organizational, hierarchical structure (see: Helland 2005). In the qualitative analysis of forum posts, three conclusions emerge from the collected material. First, the category of "algorithm" in the forum discussions is treated as a source domain, that is the usage of the concept of "algorithm" aims to explain and understand other, more abstract categories. Second, algorithms and the algorithmization of society are connected to the trends of personalization and individualization of religion and faith. Third, algorithms that also begin appearing in religious contexts are perceived as ideological objects.

## 3. Results

### 3.1. "Algorithm" as a Source Domain

The analysis of posts appearing on Catholic internet forums allows us to conclude that the category of "algorithm" is used in discourse as a metaphor. In this context, the first important conclusion is that "algorithm" is an explanatory concept/category, not an explained one. We can refer here to the concept of metaphors by George Lakoff and Mark Johnson. They claimed that our conceptual system is mainly metaphorical and, although we do not realize it in our daily routine, we think through metaphors (Lakoff and Johnson 1980, p. 3) and metaphors build whole domains of experience. The work of conceptual metaphor is to "allow us to understand one domain of experience in terms of another" (Lakoff and Johnson 1980, p. 117). Conceptual metaphor is understood as a formal statement of any idea that is inferred from metaphorical expressions (Charteris-Black 2004, p. 15). They carry structure from the "source" conceptual domain (Y) to a "target" domain (X), so that domain X is understood in terms of domain Y, and a conceptual metaphor links an abstract and complex target domain (X as *explanandum*) with a more concrete source domain that is closer to experience (Y as *explanans*) (Fauconnier and Turner 1995, p. 183; Jäkel 2002, p. 21).

In the context of the discourse analysis that conclusion reveals that "God" and "sacrum" belong to the abstract domain, while the "algorithm" seems to be a category that is better recognised and better understood closer to human experiences and perceptions. Therefore, the ideas of forum users about algorithms are transferred to ideas about God and divine reality. The divine domain manifests itself in several aspects of discourse and imaginings about algorithms and artificial intelligence, creating a broader map.

### 3.1.1. Way of Doing Things

The category of "algorithm" is used to describe a certain way of doing things or even living. An algorithm is, on the one hand, a way of life, on the other, an instruction for life. For example, there is talk of an "algorithm for the life of a Catholic."

> Life is an art. Therefore, there is not and will never be a rigid algorithm that provides a universal answer to the question of how to live. Every algorithm eventually breaks down. The Decalogue is one such algorithm [anonymous (account removed)]. [W1; 25.08.2018; 182 posts in the topic][1]

> Well, that's the trouble, because life is not so simple to give an algorithm for life [W2; 7.09.2018; 96 posts]. I agree that it's impossible to create a perfect algorithm for life, yet our minds attempt to create at least sufficiently good ones. Perhaps this is why there are diverse guidelines from psychologists, therapists, and spiritual guides of various perspectives, beliefs, and religions? [W3; 7.09.2018; 96 posts]

For W3, pondering good and evil leads to the conclusion that compiling a list of good and evil deeds is difficult and dependent on the context and individual factors. Simple guidelines drawn from the commandments, such as those that speak of loving one's neighbor, become difficult to implement in the face of ever-emerging new data, ideas, opinions, and views. A user from another forum (a 36-year-old male) requested advice on how to break free from the sin of impurity, masturbation, and viewing pornography. The advice he got spoke of a specific algorithm for liberating oneself from every sin. The algorithm consists of setting himself tasks and deadlines to "endure" without the sin of impurity for a specific, increasingly long period.

### 3.1.2. Instructions for Religious/Spiritual Practices

The concept of an algorithm also applies to guidelines and instructions regarding specific religious and spiritual practices that should unfold in a more or less precisely defined manner, combining sequences of words and actions. This may involve meditation, as well as participation in collective and individual rituals. A user (a young woman) has

prepared a set of instructions to help answer the question of whether one has sinned. She was inspired by the large number of questions asked on the forum. The author writes:

"If you are in doubt whether you have sinned follow the given algorithm". [W4; 1.11.2011; 5 posts]

The prepared algorithm is based on the conditions for a good confession, as well as a reading of the Psalms and the Catechism of the Catholic Church. The author emphasizes that this is not the only possible approach, and one should not rigidly adhere to it. The algorithm is a repeatable way of action, an instruction for confronting oneself with the Word of God, and it consists of steps, such as prayer to the Holy Spirit, examination of conscience, sorrow for sins, etc. However, in case of doubts, after several iterations of the algorithm, the author refers to a conversation with a priest as the ultimate authority.

3.1.3. God as the Programmer

God and the sacred are also present within the algorithm domain. God is referred to as the Great Programmer, the creator of the software or algorithm. In this context, inquiries arise regarding randomness (if God designed or programmed everything, does room exist for contingencies?) and free will (are the decisions made by humans genuinely their own?). Such metaphors are typically not introduced by Catholics, but start gaining traction in forums as Catholic users engage in discussions that utilize categories from the source domain (algorithm, software, programmer) to elucidate or comprehend the target domain (God, sacred). From this perspective, God, the sacred, faith, and religion emerge as more abstract phenomena, rendered more accessible to explanation and understanding through a conceptual framework that draws upon more tangible and concrete elements. An example is as follows:

The basis of the operation of this universe is certain laws and values that can be expressed mathematically... Thus, God is a mathematician and creator of the mathematics of this universe, a sort of programmer. This analogy is not perfect, but it is sometimes useful. A programmer who created a game with self-aware beings creating their games has a good level. [W3; 20.05.2019; 3215 posts]

Consequently, within the domain of an algorithm projected onto reality, understood religiously, other elements and aspects of this worldview begin to fit as well. If God is a programmer, then the creation is software, and the soul may be an algorithm:

A precise definition of (the soul) is not necessary for me, and I can provisionally agree, for example, with the concept that my soul is God's idea of me! Like a recipe, an algorithm known only to Him! Something of which the genetic code is just a physical element. (And with favourable arrangements, He will record it in the Book of Life!) [W5; 21.10.2007; 138 posts]

Using programmatic terms and concepts leads to discussions on the appropriateness of metaphors, but also the cognitive consequences of their use. Metaphors act like a pair of glasses through which one views an observed phenomenon—the glasses allow one to see a problem in a different way, to focus on a certain aspect, but at the same time they cover up others (Jäkel 2002, p. 22; Maasen 1995, p. 14). A woman, who proposed the metaphor of God as a programmer (and agreed that the metaphor is not perfect) asks about determinism and free will:

The question is whether he created the basic algorithm and let it run on its own or interfered directly with the game? Did he know what would come of it or is he as curious as we are about the results? To what extent can we project our own characteristics onto God? Or is God a mathematical formula? [W3; 20.05.2019; 3215 posts]

It becomes necessary in this situation to search for answers to questions that are not new but are renewed in the face of the metaphors used. They are questions about God's

omnipotence and attempts to link his omniscience and responsibility. If the God programmer creates software, he should know whether it is good or defective. Rationalizations such as the following appear:

> Since my actions are a function of software and conditions, the Creator is completely responsible for them anyway. [K1; 14.10.2014; 1168 posts]

> So what if god writes a random algorithm if he knows the future and what that algorithm will generate? [K2; 15.10.2014; 1168 posts]

> Even today, programmers can create a completely random algorithm, where the choice of a given option is completely random and independent of the programmer... Compare it to the capabilities of God. Can't you imagine that? [...] Knowing what will happen is not equivalent to action—God knows what will happen, but does not make it happen (man has free will). [K3; 15.10.2014; 1168 posts]

The metaphor leads to more complicated doubts and paradoxes. For example, users do not know or do not agree on how to fit the figure of Christ into the interpretation. Christ is, from this perspective, both a programmer and a program equipped with an algorithm; he has divine and human programming, or has the same "programming" (human nature) as biblical Adam. There is also an emerging interpretation that God was aware that he had created weak software containing bugs, and he allowed the software to crucify him for these bugs:

> God is the creator, the creator who refined us from the dust of the earth... That said, let's say, you wrote poor software. You won't change it, even though you know what bugs it contains? [K4; 14.10.2014; 1168 posts]

> And for this poor "software", he gave himself crucified to this software yes? [K3; 14.10.2014; 1168 posts]

This and the following quote are votes in favor of rejecting such metaphors. Users say that if we take the approach that God created the program and had no control over it, then consequently one would have to assume that:

> If Jesus had a random algorithm, God would risk a lot, if Satan succeeded in tempting Jesus, and then Satan would win. [K5; 15.10.2014; 1168 posts]

On Catholic forums, there are discussions about evolution in the context of religion. Evolution is usually contrasted with the concept of intelligent design. "Algorithm" is used in both theories. Proponents of the theory of evolution believe that evolution is a "mere" algorithm that operates not only in biology but does not have to have a clearly defined beginning. It is a process that involves small changes aimed at matching the functioning of a given entity to the environment and, as such, also occurs in economics, for example. In this view, an algorithm is rather a model of a process that can be discovered, applied, and simulated, but cannot be invented, because it is an entity that is independent and exists objectively. In contrast, Catholic debaters say that an algorithm/program requires a programmer. Although the division for atheists–supporters of the theory of evolution and Catholics–supporters of the concept of "intelligent design" is a simplification, it is present in the context of programming metaphors. The discussion is about whether the world was created by God (by the programmer), or whether it could have arisen from nothing (as atheists want). According to Catholics, writing an algorithm, even if we reduce evolution to that, requires a purpose, a plan, and a reason, while activating it requires only energy and an environment.

Another question, about the observable universe as a computer game, seems to be inspired by some science fiction movies (like *The Matrix*) and popular science publications:

> I have noticed that more and more physical scientists are getting into it. This has its consequences. If there is such a complicated game with fixed rules, there must be a Programmer. Even if the programs create themselves, someone has to create

the computer environment, and this is a big job for an intelligent genius. Even if the environment creates itself, there must be a super-intelligence that naturally also arose and created a world like ours. It's a closed circle where there is always a super-intelligence that embraces it all. [W6; 20.05.2019; 3215 posts]

### 3.1.4. Unconscious Actions

Although the very concept of an algorithm refers to the categories of rationality (rationalization), logical thinking, planning, and calculation, in discussions on the forums it often appears in the context of automatic and, therefore, unconscious behavior and actions.

> The unconscious is guided by algorithms. The instincts of animals and the unconscious reactions of humans are precisely algorithms of behaviour. A completely objective perception of the world, where there is no room for meaning, value and desire, does not provide a rational basis for any algorithm, because there is no need for any activity. [W7; 23.07.2007; 185 posts]

> A robot cannot worship since this kind of act is only afforded to persons; at most, it can be designed to mimic the gestures themselves, without, of course, any awareness of what it is doing. [And then some other user adds:] It is not a human being but an unconscious algorithm. [W8; 20.08.2007; 154 posts]

> The fact that there is such a thing as consciousness proves that God is Someone more than a manufacturer of devices. Our devices—computers do not have consciousness. Even though a computer wins against a chess champion, it will not know it, and even though it passes the Turing test, it will not have consciousness. (...) The automaton manifests a short loop: stimulus, algorithm and result. [K5; 15.08.2019; 276 posts]

The algorithm metaphor, in the context of discussions on rational actions and unconscious behavior, usually leads to discussions on free will and morality. Conceptualizations of morality as algorithm-based, i.e., unconscious behaviors and actions, emerge. As one user claims, the choice of good rather than evil is a result of the fact that humans are controlled by an algorithm or program, and humans themselves are robots who would rather choose the good option. This approach raises objections because subordination to an algorithm, the algorithmization of choices, excludes, according to other users, the freedom of choice on which morality is based. The choice to be "good" appears to be an individual matter for each person, not the implementation of a hidden agenda, regardless of whether its author is God, nature, or society. It seems that sometimes discussants operate with different meanings and metaphorize the concept of an algorithm to different degrees. It is objectionable to equate humans with robots, but the concept of an algorithm or program can be taken as a metaphor for "education" or "socialization". As one user puts it:

> And what else does "being good" mean other than being programmed to think in a certain way, that is, to make decisions in a certain way, according to certain criteria? (...) We always choose by doing some kind of profit and loss analysis. That is, we act according to a learned program. In some programs, the factor of making ourselves happy prevails and in others the "moral" factor. Either way, this is also part of the automatism of thinking. For what else is "morality" but an appropriate algorithm implemented by the brain? [W9; 5.02.2015; 41 posts]

The concept of an algorithm is also used to describe the rational, efficiency-oriented actions of an organized society, which might be the impact of religion and faith. As users claim, our orderly society (or societies) works in a way that people perform many activities automatically, without being aware of them. That is the way to save time and energy, but also the way to lose creativity, innovation, and spontaneity. The bureaucratization of life leads to "conformist nihilism" and submitting to the will of the "non-personal god of bureaucracy". Users of the Katolik.pl platform believe that there is a danger in recognizing that these laws concerning the bureaucratic organization of life have always existed, just as

atheists say about the eternal laws of physics. This is admitting that "the God–Creator is not necessary for anything" [K6; 16.09.2018; 58 posts], which applies to both the natural world and the social world.

*3.2. Individualization and Personalization*

The trend of algorithmization manifests itself in both the individualization of choices and actions and the massification of behavior. Kazimierz Krzysztofek (2014) writes about the paradox of contemporary society, in which two imperatives coexist: an emphasis on innovation and, on the other hand, on predictability. Algorithms are important for both.

Individualization, as one of the crucial categories of theories by Anthony Giddens (1991), Zygmunt Bauman (1992), or Beck and Beck-Gernsheim (2002), involves expanding the scope of opportunities to make one's own decisions and choices regardless of the structural and class pressures. The emergence of alternative lifestyles and the freedom from traditional religious institutions is indicated. With regard to religion, James Beckford (2003, p. 210) notes that what characterizes religion in late-modern societies "is the freedom of individuals to make choices about the kind of religious beliefs, [...] the ways in which individuals choose to express their religious beliefs and to use their religion". Individualization is the key process in religious change, but—according to Beckford—this statement requires consideration of the context in which the individual choices are made. Among these contexts, there are new information technologies, and—as Beckford states—"individual choices are directly or indirectly influenced by events in cyberspace" (2003, pp. 211–12). As the individualization of religion is the process of religious change, which is influenced by new technologies, the personalization of contents and services is the second side of this trend. As the process of creating unique solutions tailored to the needs of individual consumers, personalization mainly concerns marketing and sales, software, and new media, but the range of applications of the concept is expanding (Bardakci and Whitelock 2003; Vesanen 2007).

Individual religiosity, the selection of elements from the package of religion that suit a person, is combined with trend matching these elements to the user. From the forum discussions on interactions with ChatGPT, it is clear that the personalization trend is becoming real, with both positive and negative facets. There are several dimensions of such religious personalization.

### 3.2.1. Replacing Spiritual Guides

Users argue that algorithms and well-trained AI models equipped with extensive databases, including the Bible, various interpretations of the Holy Scriptures, Christian publications, guides on spirituality, Catholic forums, social media posts, and canon law, have the potential to offer guidance on matters of spirituality, faith, and canon law. This transition is anticipated to commence by replacing psychologists and psychotherapists initially, eventually extending to include "spiritual guides", such as priests.

Participating in discussions with highly trained and consistently empathetic artificial intelligence is anticipated to hold greater appeal for a generation accustomed to online communication than traditional interactions with other humans. This incremental transition towards AI-based support is regarded as a transformative advancement in how individuals approach and obtain guidance in matters related to spirituality and faith.

One of the users of Katolik.pl is afraid of the influence of language models and algorithms mostly on believers, for which:

> Instead of seeking answers in the Word of God and conversations with people, it may seem easier and safer to turn to an AI model with one's spiritual problem, hoping that it will find a satisfying "divine" solution. [K11; 26.04.2023; 58 posts]

### 3.2.2. Artificial Prayers

Conversations about interacting with ChatGPT have become a pivotal topic in forum discussions, particularly within the broader context of the influence of algorithms and

artificial intelligence on religion. This focus extends to the potential for generating religious texts, such as sacred scriptures, prayers, or sermons, as well as the perceived threats posed to religion. Users on these forums often share examples of their works created using ChatGPT and, notably, the majority of them identify as non-Catholic, including nonbelievers and Protestants.

One noteworthy user, who employed ChatGPT to compose a sermon and a prayer, made it clear that they are not necessarily defending the program, but express confidence that "'spirituality' could be created today by the algorithm". Contrasting this perspective, Catholics, as highlighted by forum participants, generally adopt a more conservative stance. According to their viewpoint, prayers generated by algorithms lack context, spiritual experience, and intention. They argue that these prayers are not genuine "prayers from the heart" but are artificial because they are "generated" rather than written. They are for "using", not for praying. Critics in the Catholic community suggest that such false prayers and other pseudo-spiritual texts may not contribute to building and strengthening spirituality. Instead, they could lead to empty devotional or even magical practices, representing what some consider a sin against the Holy Spirit. As one user of the forum Katolik.pl puts it:

> Prayers from prayer books often have a specific and known context of origin; these are approved prayers for everyone, and the purpose for which a person prays a chosen prayer can help determine whether the person is praying from the heart or not. We have the right to expect that these prayers follow the teachings of the Church and do not contain doubtful elements that could lead us astray. I know that the Church stands behind them, and supports them with its authority, and I can expect that these prayers are pleasing to God. If I pray the Litany of Loreto for the healing of a close person (the experience of the sick person and mine are intertwined in the story of life), it will likely be a prayer "from the heart" rather than a prayer supported by an algorithm, whose reference is unknown, and which has not passed through the Church's verification or even the filter of my knowledge. As you have checked, when the algorithm doesn't know what to write, it fabricates and creates the appearance of truth. [K7; 29.04.2023; 58 posts]

The ability to craft personalized prayers or devotional texts using algorithms introduces a sense of individualization, but it also opens the door for ChatGPT (or language models) to learn and potentially manipulate language. According to Catholic users, well-trained algorithms could be employed in shaping "the new man" and the new Church, aligning with the prophecy of St. Paul. According to this perspective, the envisioned Church will outwardly resemble Christianity, with preachers advocating for peace and unity. However, it is believed that this unity will be based on false foundations, resulting in a Church that is aesthetically pleasing but lacks depth, akin to the texts currently generated by AI. The threat that the Church will be turned, with the use of algorithms, into an empty ideology is emphasized. It is noticed that ChatGPT is still only "the language algorithm" not AI, but in this respect the future might be stranger than we could imagine. There are also fears that algorithms and AI will be used for the other secret and dangerous humanity goals, and expectations that the role of the Roman Catholic Church is to act as a brake during the process.

### 3.2.3. Signs and Hints from the Algorithm

In the Catholic forums, one can find stories about relinquishing decisions to algorithms. The algorithms on platforms like YouTube or other social media sites are perceived as entities capable of offering signs of fate or guidance from God. A short example concerns a user of the forum ZChrystusem [ZC1], who expressed frustration about difficulties in organizing his life. Desiring marriage, he faced challenges in finding a suitable life partner due to stringent criteria. Despite numerous attempts and setbacks, his perseverance waned, leading to a loss of motivation and hope.

One day, he decided to keep his computer running after finishing work in the office. Instead of shutting it down, he initiated a YouTube video with a sermon set to autoplay.

Leaving YouTube on overnight, he assumed that the algorithm, following its operational logic, would seamlessly transition from one video to another. When he returned to work the next day, the algorithm had led him to an intriguing sermon on marriage. A renowned preacher emphasized the importance of addressing family matters and encouraged the pursuit of a life partner.

> When the next day I checked YouTube's algorithm, it led me to a speech by Father K[...] about marriage, emphasizing the importance of not neglecting these matters and making an effort to find a spouse. [ZC1; 23.10.2019; 39 posts]

Other users frequently highlighted that the algorithm often recommended cheerful, uplifting content, restoring their mood and faith.

### 3.3. Algorithms as an Ideological Object

Algorithms are perceived as ideologically entangled, non-neutral objects. Therefore, their use by religious individuals, particularly for religious purposes, is considered problematic. A common example of such an algorithmic innovation in 2023 is the language model, ChatGPT.

Users express concern that what ChatGPT is generating nowadays is just the beginning and its capabilities are steadily growing. Soon, texts written by AI could be indistinguishable from those written by humans, such as preachers. The ongoing discussion leads to further conclusions. First, currently, priests do not always deliver their sermons, as much depends on their delivery style. Some priests may not be adept at composing sermons but can effectively "act" and deliver sermons prepared by others or generated by an algorithm. Second, although ChatGPT-4 is still in the early stages of development, it has been released to learn and, after this stage, some of its features will not be available for a fee. People are already asking ChatGPT to generate sermons, prayers, and answers to questions about spirituality or religion, contributing to the improvement of such algorithms and the creation of an enhanced version of the machine. In the future, these algorithms may create illusions for internet users using AI. Consequently, ChatGPT becomes the beneficiary of actions taken by internet users. Third, it is essential to acknowledge that there are people behind every algorithm, and these people carry certain ideologies. One of the forum users writes:

> I've noticed that some AI will fail to generate prayers that are in line with a given religion but not in line with the ideology of the AI creators (or maybe the AI itself concludes and trains itself that it's in its interest not to undermine/undermine the ideology of the creators of AI, otherwise it may be disabled. [K7; 19.07.2023; 58 posts]

Forum users are convinced that algorithms demonstrate creativity in generating texts that align with the expectations of both their audience and the creators.

Enthusiastic users of ChatGPT-4 in forums share their AI orders, presenting sermons, prayers, or answers (or whole chats) generated by the algorithm. These users are typically not Catholics, yet they find themselves on Catholic forums. In response, Catholics take on the role of "cautious" or "cold sceptics", highlighting inconsistencies or deficiencies in the generated texts, such as sermons resembling those written by an elementary school student or prayers lacking emotional depth. Ideological biases are also pointed out, with the common accusation that some prayers or sermons are "too Protestant" and not in line with the spirit of Catholicism. For example, users criticize ChatGPT-generated content for emphasizing "the new life" and excessive thanksgiving, which—they argue—aligns more with Protestant thinking, than with deep Catholicism. Some users also note unnatural phrases or inconsistencies with tradition, such as when the algorithm addresses Jesus as the Father in such a prayer. On the other hand, a Protestant, who is a visitor to a Catholic forum, posts examples of his conversations with ChatGPT on religious topics. When Catholic discussants begin to express impatience with his enthusiasm for ChatGPT and encourage him to engage in relationships with other people or to read worthwhile books,

he expresses satisfaction that ChatGPT does not provide answers that are too Catholic, unlike the recommendations from other authors:

> By the way—isn't it a cold shower for you to realize that you don't want to get rich with what people for whom God was close to have reached, and you delight in the instructions of artificial intelligence? [DM1; 20.04.2023; 121 posts]

> I won't be happy, if [ChatGPT] gives me purely Catholic content; fortunately, it almost doesn't (and the people you recommend notoriously do—so I know what to choose!). [DM2; 20.04.2023; 121 posts]

The algorithms on the most popular social media platforms, such as Facebook and YouTube, are also influenced by ideologies hostile to the Roman Catholic Church. Users argue that these algorithms operate on a social credit system, where the number of likes, subscriptions, and shares influences their behavior. This system, combined with the propaganda of the world's elites, tends to elevate content representing a specific ideology over alternative viewpoints, contributing to the "grooming" of citizens. In the users' opinions, Google and Facebook algorithms are often perceived as "leftist". These algorithms, which promote specific content and block inappropriate material, have effectively replaced traditional censorship. Additionally, social media users engage in self-censorship. One user observes:

> In the case of BigTechs and platforms like Facebook, what is allowed to be written and what is not is determined by an algorithm, or rather "the leftist behind it", who authoritatively decides that, for example, Christian content violates certain standards, while insulting Christians does not. [K8; 31.01.2021; 252 posts]

And another user adds:

> Regarding algorithms, they are written by people (often teams, not just one leftist) and reflect the policies of the management of the company or platform where they are created. [K9; 2.02.2021; 252 posts]

It is pointed out that social media platforms have gone from being harmless tools that facilitate communication to becoming manipulative machines that pose a threat to society. This is due to algorithms that make people addicted and worse. As one user writes:

> Unfortunately, any tool can serve evil and good. However, there are specialized tools. Some tools serve mainly evil. Or mainly good. [...] Facebook is used by some for good purposes. What percentage of information sent via Facebook has a salutary effect? And what, the opposite? What percentage is mainly a waste of time? The time, that is God's gift. [ZC2; 13.03.2022; 94 posts]

## 4. Discussion

Analysis of the meanings attributed to algorithms on Catholic forums reveals, above all, a wide variety of approaches. There is no single understanding of an algorithm, and it is not possible to provide a uniform definition that leans more towards a "mathematical" understanding, or an approach that treats algorithms as complex socio-technological systems. It can be assumed that, very often, the use of the term "algorithm" does not involve its explicit conceptualization. Meanings circulate around the forums and are referred to contextually. Importantly, unlike the Church's position and the discussions held by theologians, forum users more often reference their own experiences and frequently speak of algorithms as something concrete, which directly affects their actions or that they can use. They are less likely to discuss the far-reaching consequences of algorithms or normative declarations, stating that both algorithms and artificial intelligence should act ethically, for human dignity, or for social justice.

There is no consistent vernacular theory of algorithms, as there is not even one definition. When discussing the understanding of algorithms, there is a wide scope of meanings to consider, starting from a strictly mathematical understanding (they were not the object

of this study). The concept of an "algorithm" is often used metaphorically and serves, according to modern concepts of metaphor, to describe and understand more complex and abstract concepts concerning theological teachings about God, the sacred, salvation, free will, but also evolution. This shows that, paradoxically, the concept of an "algorithm" seems to be more comprehensible, perhaps only intuitively, and closer to everyday experience than issues of faith, religion, and the sacred, to which Catholic forums are devoted. There are quite simple metaphors referring to ways of doing things or instructions for devotional practices, but also more sophisticated ones presenting God as the programmer, with the consequence that the world is a program. On the other hand, algorithms were understood as a device/machine that are actually working regardless of people and, even more so, are influencing their thoughts and behaviors. Regardless of the different ways of understanding, algorithms in the context of religion are "the black box", a system whose workings are mysterious and controversial. Although the "algorithm" is perceived as an "enigmatic technology", there are attempts to dismantle the "black box", to dissect it and reveal that there is always a person or a team of people behind the code. This process involves "uncovering" the ideological base that, in turn, guides the code makers.

Algorithms, in the context of religion, problematize the issues related to agency and causality, as they are imagined as devices that work independently of human will. As Massimo Airoldi claims, algorithms are a system or machine, but are seen first and foremost as the output of their work. The principles of operation and mechanism remain unknown. Airoldi notes that there are many pieces of research on the controlling and influencing of citizens by the output of algorithms (Airoldi 2022, p. 17). As one of the forum users pointed out, programmers are able to create algorithms, which operate and make decision that cannot be predicted by the authors. The attitude of the Catholic users of the religious forums seems to be marked with the awareness of that possibility. The social imagination of algorithms leads to predictions that algorithms will be able to create spirituality, replace priests and other spiritual guides, and give spiritual advice.

In "folk theories", the algorithm is attributed with a causal force or is considered a causal force, exerting increasing pressure on society. In this sense, it is perceived as a threat to religion, faith, and the Church, rather than as an ally. Indeed, fragmented decision-making and dispersed causality can undermine the perception of divine causality. The emergence of ChatGPT-4 and other tools and algorithms that constitute artificial intelligence imaginary reveals a new dimension of reality *in statu nascendi*. The process of creating an algorithmized reality is underway, and people of faith, including Catholics, are observing it with a mixture of concern and fascination. There is no shortage of skeptical voices, such as [K10], who asserts, "We live in dark times, and if we also reduce civilization to aberration and corruption (social engineering is working at full speed), then God will show us no mercy. Nature is on God's side and not ours, so when we violate God's laws, we will suffer the consequences".

Discussions about algorithms raise questions about power and authority. In the opinion of users of the studied forums, one of the dangers for social life and for religion is the algorithmization of society, which is close to the notion of bureaucracy, as bureaucracy makes society work, according to eternal-like laws similar to the eternal laws of nature proclaimed by atheistic science. Referring to an algorithm as a system (not to metaphorical meanings), it is highlighted that it may, by manipulating language and knowledge, play a significant role in creating new men and new religious communities hostile to the Church. Personalized content "generated" by technology, i.e., ChatGPT, is often recognized on the forum as connected with some ideology (usually leftist) or another denomination (usually Protestant). The meanings ascribed to algorithms present them, according to these particular, qualitative data, as specific and tangible (even when they are used to build certain metaphors) objects equipped with causality and power, which arouse cautious and suspicious attitudes, as the algorithmization of society is perceived as the thread to faith, religion, and the Church.

The three main conclusions presented in the Results Section do not exhaust the subject; they allow us to explore the primary themes and tropes emerging in Catholic images about algorithms. An analysis of posts appearing on online forums is not an ideal method for building a precise typology or classification of folk theories about algorithms. Nevertheless, it provides insights into the directions of thinking about the meanings, power, and causality of algorithms from the perspective of forum users, where—by definition—topics related to faith and spirituality are explored. Since the pace of the algorithmization of society and the development of artificial intelligence cannot be halted, topics concerning their meanings and role in religion will gain popularity, prompting reflections not only from theologians, but also from "ordinary" believers. Certainly, in-depth research will be needed on the understanding of algorithms in the context of religion.

**Funding:** This research received no external funding.

**Data Availability Statement:** No new data were created or analyzed in this study. Data sharing is not applicable to this article.

**Conflicts of Interest:** The author declares no conflicts of interest.

## Note

1     Quotes from forum posts were coded concerning specific forums: W—forum Wiara.pl; K—forum Dyskusje Katolik.pl; ZC—forum ZChrystusem; DM—forum Dolina Modlitwy. The date the post was published and the number of posts on the topic were added.

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
