# Peer review of "Algorithms and Faith: The Meaning, Power, and Causality of Algorithms in Catholic Online Discourse"

_religions, doi:10.3390/rel15040431_

Round 1

Reviewer 1 Report

Comments and Suggestions for Authors

The article delves into a current and compelling topic - algorithms and their perception - to elucidate prevalent ideas and approaches within the religious context in Poland.

However, I believe the proposed study requires further investigation and clarification, which I will discuss below.

The author(s) should better justify their decision to analyse Catholic forums rather than social media. I'm not sure if it makes sense, given that we're talking about grassroots and vernacular ideas and concepts, to limit the field to the analysis of forum comments in the age of ubiquitous media like social media. It would be useful to understand the scale of the phenomenon: how many people participate in the debates in the analysed forums? Given the fact that more than 70% of Polish use social media (source: Statista), it is necessary to explain why the forums were chosen for analysis. I believe it is critical to grasp the importance of such online discussion spaces in the Polish national debate, particularly for foreign readers.

Referring to the conducted analysis, some additions to the methodology are necessary. The number of topics analyzed (in which forums are structured) are indicated, but we don’t know how many posts have been analyzed, how many unique users are involved, and when the contents have been published. I think that it is important, because too often scholars on religion and online focus their efforts to analyze too very small phenomena.

Regarding the categories of meanings assigned to the algorithms, I suggest to better define the second one category (“individualization and personalization”), to make it more consistent with the other two (“algorithm as a source domain” and “algorithms as ideological devices”). Finally, pay attention to some small typos in the text, in particular with regard to the numbering of paragraphs and sub-paragraphs.  

I would suggest in the future to consider giving interviews to those who publish contents on the subject under investigation to deepen their interpretation of the phenomenon considered.

Author Response

Thank You for Your review, please see the attachment

Reviewer 2 Report

Comments and Suggestions for Authors

I find the topic very up tp date, and well discoused. Citing from recent published sources is very important. Although there are some in-text citation where the author mentioned the page of the source while in other sources the author didn´t mention the page number, for example: (Willson, 2017) in the introduction.

I find the introduction a bit long, maybe some parts can be divided under different subtitles.

In  my opinion, it might be better to put the "Materials and Methods" part before the results, so the reader can understand the result section better.

Author Response

Thank You for Your review, please see the attachment. Kind regards.

Reviewer 3 Report

Comments and Suggestions for Authors

The topic of the article is very interesting and the study's results are presented in a logical manner, and the argumentation is easy to follow. Except for some typos ("Polich" on p. 5 and "polish" in the Abstract) and punctuation which need to be edited, the paper requires a few modifications.

First of all, while the research question is interesting, it's descriptive - while the Author makes a thorough review of relevant literature, it seems she/he doesn't problematize the results in light of the theoretical background. There is some mention of theoretical consequences in the Discussion, but it's very brief and doesn't offer the Reader any revision of the existing concepts. It's also unclear how the results correspond with the said literature. Overall, while the theoretical foundations are solid, and the research itself well presented and argumented, it often seems like they are two separate entities. To mitigate this issue, I suggest 1) a revision of the research problem and question so that they are problem-based, and 2) a more in-depth reflection on the addition to theory in the Discussion. 

Comments on the Quality of English Language

The paper needs proofreading: some typos and punctuation need revising (see my comment above).

Author Response

(The authors gave the same response as above.)

Round 2

Reviewer 1 Report

Comments and Suggestions for Authors

I appreciated the revisions done by the author. In particular, the reasons for examining Catholic forums instead of social media were appreciated by me. I suggest to delete the phrase “The most important reason for choosing internet forums rather than social media was that, there were more discussions about algorithms there.” (p. 6 rows 293-294) because the previous explanations are sufficient.

As well, I appreciated more information on the description of the forums that were analyzed.

However, I still have some concerns regarding the overall paper.

-          I believe the paper is overly descriptive, as stated in the abstract: "The purpose of the article is to present grassroots concepts and ideas about "the algorithm" in the religious context". It requires a more interpretative approach, after better defining the theoretical framework. It is necessary to clarify the research questions or hypotheses.

-          There is a confusion between algorithms in general and ChatGPT (in some cases, the author specifies ChatGPT-4). It’s possible that the confusion stems from analysing an overly long time period (from 2004 to 2023). That means for example that up to a certain point it’s true what the author writes at p. 5 righe 226-227 "the religious" understanding may differ from the definition provided by the programists and computer scientists.", but it’s not fully true if we consider ChatGPT and other artificial intelligence in which conditioning is due to also by the internet users’ ideas, ideologies and religious convictions who use AI tools.

-          The analysis conducted brings out theological issues that call religious authority into question. In my opinion, what is missing is an analysis of the consequences of the spread of algorithms and AI for a complex institution like the Catholic Church. For example, in case of “Instructions for religious/spiritual practices”: what consequences for the Catholic authority? Maybe, the author would clarify what is the approach used in the analysis (theological or sociological analysis?) in order to better focus the work done, defining better the used theoretical framework. In my opinion, too many concepts and theories only mentioned (e.g. deep mediatization).

-          In my opinion, in order to avoid the analysis requiring theological skills (if I have not misunderstood this is not the purpose of the paper), every time the author compares God or "divine agency" with algorithms he must make a choice: What is the reading key? I suggest to refer to the consequences for religious authority, consulting the existing literature on the changes and challenges posed by technological innovations to institutions and in particular to religious institutions.

-          P. 13 row 616: on the basis of what motivations, the author states that “the role of the Roman Catholic Church is to act as a brake”? I would avoid expressing judgments on what the Church should or should not do.

Minor suggestions:

-          I suggest to insert almost one excerpt for each identified category and add data in which the content has been published.

-          P. 12 row 606: ChatGPT is not a device.

-          P. 13 row 614: too reductive to affirm that “ChatGPT is still a simple language algorithm”.

-          P. 9 row 440: there is a typo.

Author Response

Thank you again for Your time and your comments. I am sending my responses in the attachment. 

Reviewer 3 Report

Comments and Suggestions for Authors

Thank you for revising the article and including my suggestion. I think overall it reads better now, and the objectives of the article are clearly stated. There was some improvement to the main body and the conclusions which I think are sufficient for the article to be published.

Author Response

Thank you again, for your comments.